# Research and Experimental Analysis of Hydraulic Cylinder Position Control Mechanism Based on Pressure Detection

**Rulin Zhou [1,*], Lingyu Meng [1,2], Xiaoming Yuan [3] and Zishi Qiao [1]**

1   Beijing Tianma Intelligent Control Technology Co., Ltd., Beijing 100013, China; mengly@tdmarco.com (L.M.); Qiaozs@tdmarco.com (Z.Q.)
2   School of Mechanical Electronic & Information Engineering, China University of Mining & Technology, Beijing 100083, China
3   Hebei Provincial Key Laboratory of Heavy Machinery Fluid Power Transmission and Control, Yanshan University, Qinhuangdao 066004, China; xiaomingbingbing@163.com
*   Correspondence: zhourl@tdmarco.com

**Abstract:** This paper studies the precise position control of the hydraulic cylinder in the hydraulic support. The aim of this paper is to develop a method of hydraulic cylinder position control based on pressure and flow coupling, which takes the coupling feedback of load and flow into account, especially in the scene of cooperative control under the condition of multiple actuators and variable load. This method solves the problems of slow movement and sliding effect of hydraulic support in the traditional time-dependent hydraulic position control, as well as better realizes the intelligent and unmanned development of the fully mechanized mining face. First, based on the flow continuity equation and Newton Euler dynamic equation, the flow and stroke control model with the input and output pressure of hydraulic cylinder is established. Then, the effectiveness and correctness of the control model are verified by the comparison between the hydraulic system simulation software, AMESim, and the experiment. Finally, a test system is built. When the system pressure is large than 10 MPa, the error between the data determined by the fitting algorithm and the actual detection data is within 5%, which verifies the effectiveness of the theory and simulation model.

**Keywords:** hydraulic support; position control; pressure detection; flow integration; switch control

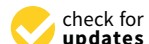



## 1. Introduction

Electro-hydraulic position servo system is a system that precisely controls the output displacement of the actuator and has been widely used in aerospace, metallurgy, engineering machinery, and other fields [1–3]. Based on the difference measured by the displacement sensor, and the target displacement to realize the position accuracy of the actuator, is the mainstream position control method to control the high-precision and high response hydraulic valve [4–7]. This control method is commonly used in variable displacement control of pump station, fault diagnosis [8,9], servo valve [10,11], and hydraulic cylinder control [12].

At present, in some scenarios requiring high precision control at home and abroad, the precise position control of hydraulic cylinder is mainly controlled by proportional valve or servo valve [13–15]. Lan Li presented a novel nonlinear model and high-precision lifting motion control method of a hydraulic manipulator driven by a proportional valve [16]. Luyue Yin proposed a novel finite-time output feedback controller with parameter adaptation for electro-hydraulic servo systems [17]. However, because of low energy efficiency and high cost of these control valves, the application scenarios are limited [18]. In recent years, many scholars have carried out new explorations in order to reduce the cost of position control system valves and improve the application scope. For example, Emeibo has created an integrated digital hydraulic actuator, which integrates the feedback and control of valve and cylinder into a whole to realize the precise control of displacement [19], and Zhou Chuanghui proposed a high-precision idea of hydraulic cylinder output position through

a two-stage displacement closed-loop feedback control with ordinary electro-hydraulic control directional valve and hydraulic throttling damping as the core components [20]. Jin liyang of Zhejiang University proposed an precise control system of hydraulic cylinder based on a electro-hydraulic control directional valve, mainly from the perspective of control algorithm and model identification [21], and Jinbo et al. proposed an electro-hydraulic position control system using a large flow switch valve and small flow proportional valve in parallel to replace large flow servo valve [22]. Jiang Haiyu studied the electric and hydraulic position control system of a valve controlled cylinder based on a high-speed switch valve [23]. Zhao Ruihao proposed a novel two-position three-way electro-hydraulic proportional directional flow valve for hydraulic roof support [24]. Shi Jianpeng proposed a velocity and position combined control strategy based on mode switching control the boom and arm of hydraulic excavator [25]. Ahsan Saeedzadeh proposed a digital hydraulic circuit using a fast-switching on/off valve instead of servo valves to control the position of a hydraulic actuator [26].

In the coal industry, the hydraulic system of a fully mechanized mining face mainly uses a high flow switch hydraulic valve with 5% concentration emulsion as medium. Considering the restrictions of coal safety regulations on power consumption, reliability, cost of coal safety regulations, and the pollution level of the hydraulic system being low, the traditional electrohydraulic position control system cannot be applied to the hydraulic system of a fully mechanized mining face [27,28]. Generally, the length of a fully mechanized coal mining face is 300 m, arranged by more than 150 hydraulic supports in line, and each hydraulic support is composed of more than 10 valve-controlled cylinder units. The hydraulic system of a fully mechanized mining face has at least one main inlet and outlet, and it is equipped with pressure sensors. Therefore, the hydraulic system of a fully mechanized mining face is a complex hydraulic cylinder cluster control system. To follow up with shearer, the hydraulic support needs to carry out relevant action and attitude adjustment. Due to load difference, the action of different hydraulic cylinder units causes the mismatch of system flow and pressure. At the same time, due to the harsh working environment, the valve controlled hydraulic cylinder is mainly open-loop control and time control. Therefore, when the system pressure and load change, it is easy to cause the uncontrollable stroke of the hydraulic cylinder, which affects the attitude control of hydraulic support and ultimately affects the mining efficiency.

The straightness control of hydraulic support in a fully mechanized mining face is a key technology for the stable operation of an intelligent and unmanned mining face. With the introduction of foreign LASC (Longwall Automation Steering Committee) technology, the precise detection of straightness of a fully mechanized mining face has been realized, and the position error is at cm level [29,30], However, the precise control of the advancing hydraulic cylinder is mainly controlled by a large flow switch valve based on position detection. Although the logic cartridge valve with a large and small flow switch was adopted later, there is still room for improvement in the control accuracy and structure [31]. At the same time, other hydraulic cylinders of the hydraulic support do not have displacement sensors, and most of the control is mainly through the switch time control of the solenoid valve, which causes low system efficiency; on the other hand, it is easy to lose the position of the hydraulic supports.

As one of the key technologies for the normal, reliable, and efficient operation of the intelligent working face, the precise control of the hydraulic cylinder is not only the guarantee of the straightness control of the working face but also the basis for the efficient operation of the hydraulic support [32]. However, due to the special environment of the working face, only the advancing cylinders have displacement sensors, the rod-less chamber of the leg and the main liquid inlet pipeline have pressure sensors, the hydraulic valve is mainly a large diameter and large flow switch structure, and the hydraulic support has high quality large inertia, and leading to the precise control of the hydraulic cylinder in the hydraulic support has always been a difficult problem. Because of the most reliable sensor of the hydraulic system of the fully mechanized mining face is the pressure sensor,

besides it being widely used and less costly, it is the most reliable as the system input in the actual operation process. The main contribution of this paper is combining with the matching relationship between system pressure and flow. Our paper proposes an electric-hydraulic position control system based on pressure detection, which can greatly improve the efficiency of movement and sliding of hydraulic support compared with the traditional time-dependent hydraulic position control and better realize the intelligent and unmanned development of the fully mechanized mining face. With the ordinary electric-hydraulic control directional valve as the control element, the displacement of the hydraulic cylinder is analyzed and predicted through real-time pressure monitoring, so as to achieve the precise determination of multi-stage hydraulic cylinder. The system has low cost and moderate control accuracy. It is especially suitable for an application scenario without displacement sensors but still needs location control. It has certain engineering application value.

## 2. Mechanism Analysis of Hydraulic Support System

Because of the hydraulic support function principle of each sub-hydraulic system is similar, the same equivalent valve controlled cylinder model is analyzed. The principle of the hydraulic system is shown in Figure 1. When the hydraulic cylinder extends, the left directional valve 2 is energized. The liquid enters the rod-less chamber through filter1, left directional valve 2, and check valve 3. The liquid in the rod chamber flows back to the tank through right directional valve 2 and return circuit breaker 5. After the hydraulic cylinder is extended in place, the left directional valve 2 is de-energized and is returned to its original position. The check valve 3 locks the rod-less chamber, and the hydraulic cylinder position is maintained. When hydraulic cylinder retracts, the right directional valve 2 is energized. The liquid enters the rod chamber through filter1 and right directional valve 2. High pressure opens check valve 3, and liquid in the rod-less chamber flows back to the tank through check valve 3, left directional valve 2, and return circuit breaker 5.

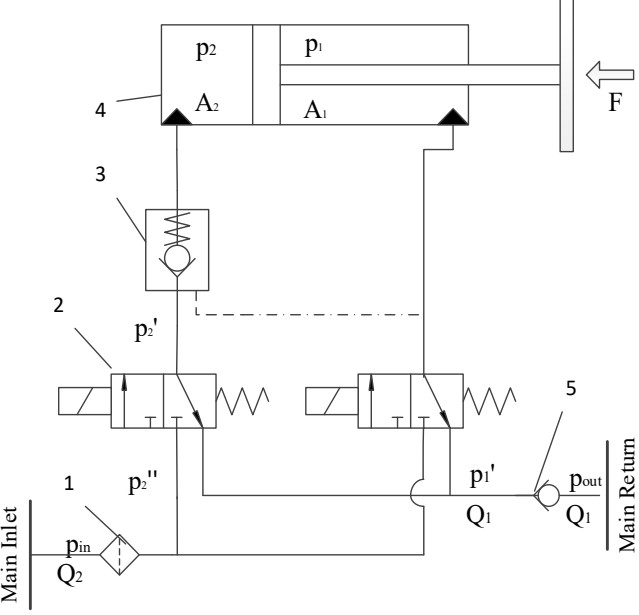

**Figure 1.** Schematic diagram of hydraulic system of hydraulic cylinder unit. 1. Filter. 2. Directional valve. 3. Liquid control check valve. 4. Hydraulic cylinder. 5. Return circuit breaker.

The hydraulic valves used in the hydraulic system of a fully mechanized mining face are a fixed opening switch valve spool structure, including a one-way valve and directional valve. The specific parameters can be fitted by flow resistance experiment and simulated by fixed throttling.

The equation of displacement change of the hydraulic cylinder at steady state is:

$$L = \int_{t_1}^{t_2} \frac{Q}{A} \, dt. \tag{1}$$

The force balance equation of rod and rod-less chambers of the hydraulic cylinder is:

$$p_2 A_2 - p_1 A_1 - F = ma. \tag{2}$$

The valve port flow equation is:

$$Q = C_V A \sqrt{\frac{2}{\rho} \Delta p} = k_p \sqrt{\Delta p}, \tag{3}$$

where $Q$ is the total output/input flow; $L$ is the displacement of the hydraulic cylinder; $A$ is the corresponding area of the hydraulic cylinder chamber w/wo rod; $t_1$, $t_2$ are the start and end time; $p_2$, $p_1$ are the main inlet and return pressure, respectively; $F$ is external joint force, and the direction is related to the direction of friction and load; $a$ is the accelerating rate of the system; $\Delta p$ is pressure difference; $C_V$ is the flow coefficient; $k_p$ is the comprehensive flow coefficient.

*2.1. Hydraulic Cylinder Extension Model*

Powered on the left working position of the solenoid valve, the following formula can be derived:

$$Q_2/Q_1 = A_2/A_1 , \tag{4}$$

$$p_{in} - p_2'' = \Delta p_2'' , \tag{5}$$

$$p_2'' - p_2' = \Delta p_2' , \tag{6}$$

$$p_2' - p_2 = \Delta p_2 , \tag{7}$$

$$Q_2 = k_2'' \sqrt{\Delta p_2''} = k_2' \sqrt{\Delta p_2'} = k_2 \sqrt{\Delta p_2}. \tag{8}$$

We combine the above formula to obtain:

$$Q_2 = k_2^x \sqrt{\Delta p_2^x}, \tag{9}$$

where $Q_2$ and $Q_1$ are unit input and output flow, respectively; $A_1$ and $A_2$ are the area of the hydraulic cylinder chamber w/wo rod, respectively; $p_{in}$ and $p_{out}$ are the main inlet and return pressure, respectively; $Q$ is the total input flow; $k_2''$ , $k_2'$, $k_2$ are the comprehensive flow coefficient of filter, the comprehensive flow coefficient of reversing valve, and the comprehensive flow coefficient of check valve, respectively; $k_2^x$ is the comprehensive flow coefficient from the rod chamber to the hydraulic valve at the main return when the hydraulic cylinder is extended (Equation (11)); $\Delta p_2''$, $\Delta p_2'$, $\Delta p_2$ are the pressure difference of filter, the inlet pressure difference of directional valve, and the inlet pressure difference of check valve, respectively; $\Delta p_2^x$ is the pressure drop from the rod chamber to the main return when the hydraulic cylinder is extended (Equation (10)).

$$\Delta p_2^x = p_{in} - p_2, \tag{10}$$

$$k_2^x = \sqrt{\frac{1}{(\frac{1}{k_2''})^2 + (\frac{1}{k_2'})^2 + (\frac{1}{k_2})^2}}. \tag{11}$$

Similarly, the other flow equation of the hydraulic cylinder can be obtained as:

$$Q_1 = k_1^x \sqrt{\Delta p_1^x}, \tag{12}$$

where $\Delta p_1^x$ is the pressure drop from the main inlet to the rod-less chamber when the hydraulic cylinder is extended. $k_1^x$ is the comprehensive measurement coefficient from the rod-less chamber to the hydraulic valve on the main inlet side when the hydraulic cylinder is extended.

$$\Delta p_1^x = p_1 - p_{out}, \tag{13}$$

$$k_1^x = \sqrt{\frac{1}{(\frac{1}{k_1'})^2 + (\frac{1}{k_1})^2}}. \tag{14}$$

System flow equations can be obtained by solving Equations (2), (4), (9) and (12):

$$Q_2 = \sqrt{(p_{in}A_2 - p_{out}A_1 - F)/[\frac{A_1^3}{(k_1^x)^2(A_2^2)} + \frac{A_2}{(k_2^x)^2}]} = k_s \cdot f_s(p_{in}, \ p_{out}), \tag{15}$$

where

$$f_s(p_{in}, \ p_{out}) = \sqrt{(p_{in}A_2 - p_{out}A_1 - F)}, \tag{16}$$

$$k_s = \sqrt{1/[\frac{A_1^3}{(k_1^x)^2(A_2^2)} + \frac{A_2}{(k_2^x)^2}]}. \tag{17}$$

The hydraulic cylinder displacement formula is shown as follows:

$$L = \int_{t_1}^{t_2} \frac{Q_2}{A_2} \, dt. \tag{18}$$

According to the pipeline layout, the pipeline flow resistance mainly includes the friction head loss and local flow resistance loss. The general Reynolds number is much larger than 2300 for hydraulic systems of a fully mechanized mining face, so it is checked according to the turbulent state [33].

The friction head loss equation is:

$$p_f = \frac{\rho}{2} \cdot \lambda \cdot \frac{l}{d} \cdot v^2 = \frac{8\rho\lambda l q^2}{\pi^2 \cdot d_f^5}, \tag{19}$$

Where, according to Ariituri formula:

$$\lambda = 0.11 \cdot \left(\frac{\Delta}{d} + \frac{68}{Re}\right)^{0.25}. \tag{20}$$

The local flow resistance loss equation is:

$$p_\xi = \frac{\rho}{2} \cdot \xi \cdot v^2 = \frac{8\rho\xi q^2}{\pi^2 \cdot d_\xi^4}, \tag{21}$$

where $\rho$ is the emulsion density, equal to 998 kg/m$^3$; $l$ is the pipe length; $d_f$ is the pipe diameter; $\lambda$ is the friction coefficient; $\xi$ is the local resistance coefficient; $\Delta$ is surface roughness, which equals to 0.05 mm; $d_\xi$ is the inner diameter of the pipe joint; $v$ is the fluid velocity.

According to the above analysis, for the hydraulic system of the hydraulic cylinder unit, during the steady-state extension of the hydraulic cylinder, the hydraulic control check valve, directional valve, and pipeline joints can all be regarded as fixed damping structures. The steady-state input flow of the system can be predicted only by detecting the inlet and outlet pressure $p_{in}$ and $p_{out}$ of the system. Finally, the position and displacement of the hydraulic cylinder can be estimated through the integration of the flow.

### 2.2. Hydraulic Cylinder Retracted Model

Powered on the right working position of solenoid valve, the steady state flow continuity equation can be obtained as:

$$Q_2/Q_1 = A_1/A_2 ,\tag{22}$$

$$Q_2 = k_2^{x_2} \sqrt{p_{in} - p_1} ,\tag{23}$$

where $k_2^{x_2}$ is the comprehensive flow coefficient from the rod chamber to the hydraulic valve at the main inlet when the hydraulic cylinder retracts.

$$k_2^{x_2} = \sqrt{\frac{1}{(\frac{1}{k_2''})^2 + \left(\frac{1}{k_1}\right)^2}} .\tag{24}$$

Similarly, the other flow equation of the hydraulic cylinder can be obtained as:

$$Q_1 = k_1^{x_1} \sqrt{p_2 - p_{out}} ,\tag{25}$$

where $k_1^{x_1}$ is the comprehensive flow coefficient from the rod-less chamber to the hydraulic valve at the main return when the hydraulic cylinder retracts.

$$k_1^{x_1} = \sqrt{\frac{1}{(\frac{1}{k_1'})^2 + (\frac{1}{k_2})^2 + (\frac{1}{k_2'})^2}} .\tag{26}$$

System flow equations can be obtained by solving Equations (2), (22), (23) and (25):

$$Q_2 = \sqrt{(p_{in} A_1 - p_{out} A_2 + F)/[\frac{A_1}{(k_2^{x_2})^2} + \frac{A_3^3}{(k_1^{x_1})^2 \cdot A_1^2}]} = k_h \cdot f_h(p_{in}, p_{out}) ,\tag{27}$$

where

$$f_h(p_{in}, p_{out}) = \sqrt{p_{in} A_1 - p_{out} A_2 + F},\tag{28}$$

$$k_h = \sqrt{1/[\frac{A_1}{(k_2^{x_2})^2} + \frac{A_3^3}{(k_1^{x_1})^2 (A_1^2)}]}.\tag{29}$$

The above analysis is based on the condition that the inlet and outlet pressures are known and detectable, the flow required for the action of a single hydraulic cylinder can be calculated, and the stroke displacement of the hydraulic cylinder can be calculated through flow integration. Similarly, when multiple hydraulic cylinder units apply in the fully mechanized mining face, the accumulated flow is the input flow of all units.

$$Q_Z = \sum_1^M Q_2^i .\tag{30}$$

## 3. Simulation and Experiment

### 3.1. Valve Control Cylinder Simulation Model

The basic working principle of the hydraulic support is to extend and retract the hydraulic cylinder through the hydraulic pump. It mainly includes three basic circuits: leg control circuit, advancing cylinder control circuit, and balance cylinder control circuit. The advancing cylinder control circuit is one of the important circuits of the hydraulic system of hydraulic support. Establishing the hydraulic simulation model of the advancing control circuit, and studying the key factors affecting the pulling and pushing of hydraulic support from the mechanism, is of great significance to improve the advancing control precision. Figure 2 shows the advancing control circuit, including advancing cylinder, hydraulic

control check valve, control signal, liquid supply circuit, etc. Table 1 is the simulation parameters of the hydraulic system of the advancing control circuit.

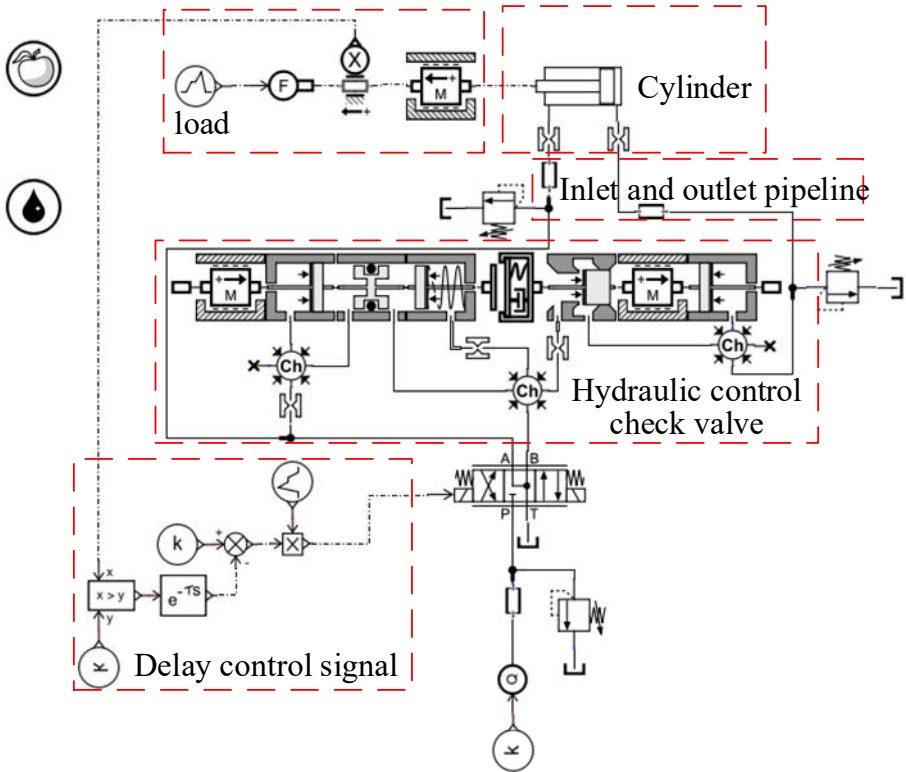

**Figure 2.** Hydraulic schematic diagram of advancing control circuit.

**Table 1.** AMESim simulation parameters of the hydraulic system of control circuit advancing.

| Symbol | The Parameter Name | Magnitude |
|---|---|---|
| $Q$ | Main System Flow/(L/min) | 400 |
| $\rho$ | The emulsion density/(kg/m$^3$) | 998 |
| $E$ | Bulk modulus of elasticity of emulsion/MPa | 2010 |
| $p_{set}$ | Set pressure of system safety valve/MPa | 31.5 |
| $D$ | Piston diameter of advancing hydraulic cylinder/mm | 180 |
| $d$ | Diameter of piston rod of advancing hydraulic cylinder/mm | 120 |
| $L$ | Stroke of advancing hydraulic cylinder/mm | 900 |
| $d_r$ | Inlet diameter of advancing hydraulic cylinder/mm | 11.2 |
| $d_c$ | Outlet diameter of advancing hydraulic cylinder/mm | 11.2 |
| $m$ | Load mass/kg | 500 |
| $F1$ | Coulomb friction of advancing hydraulic cylinder/N | 7000 |
| $F2$ | Static friction of advancing hydraulic cylinder/N | 10,000 |
| $h_f$ | Flow resistance of electro-hydraulic control directional valve/(L min$^{-1}$/MPa) | 250/5 |
| $d_f$ | Valve spool diameter of hydraulic control check valve/mm | 33 |
| $d_{\text{spool}}$ | Diameter of control piston of hydraulic control check valve/mm | 13 |
| $d_{\text{gan}}$ | Ejector rod diameter of hydraulic control check valve/mm | 7 |

### 3.2. Valve Control Cylinder Test System

In order to verify the relationship between hydraulic system parameters and hydraulic cylinder position under different working conditions, such as pressure, flow, etc., an

advancing hydraulic cylinder test platform is built. The test system takes the hydraulic system of Yujialiang 52,301 working face in Shendong as a reference. The experiment adopts 400 L/min emulsion pump as liquid supply and includes hydraulic components, such as hydraulic control check valve, electro-hydraulic control directional valve (hereinafter referred to as "directional valve"), hydraulic cylinder, safety valve, return circuit breaker, and cut-off valve in the system. The experimental schematic diagram is shown in Figure 3. The P port of the directional valve is connected to the emulsion pump, one branch of the directional valve port C1 is connected to the rod chamber of the hydraulic cylinder, the other branch is connected to the control port of the hydraulic control check valve, the directional valve port C2 is connected to the hydraulic control check valve, and the locking cavity of the hydraulic control check valve is connected to the rod-less chamber of the hydraulic cylinder.

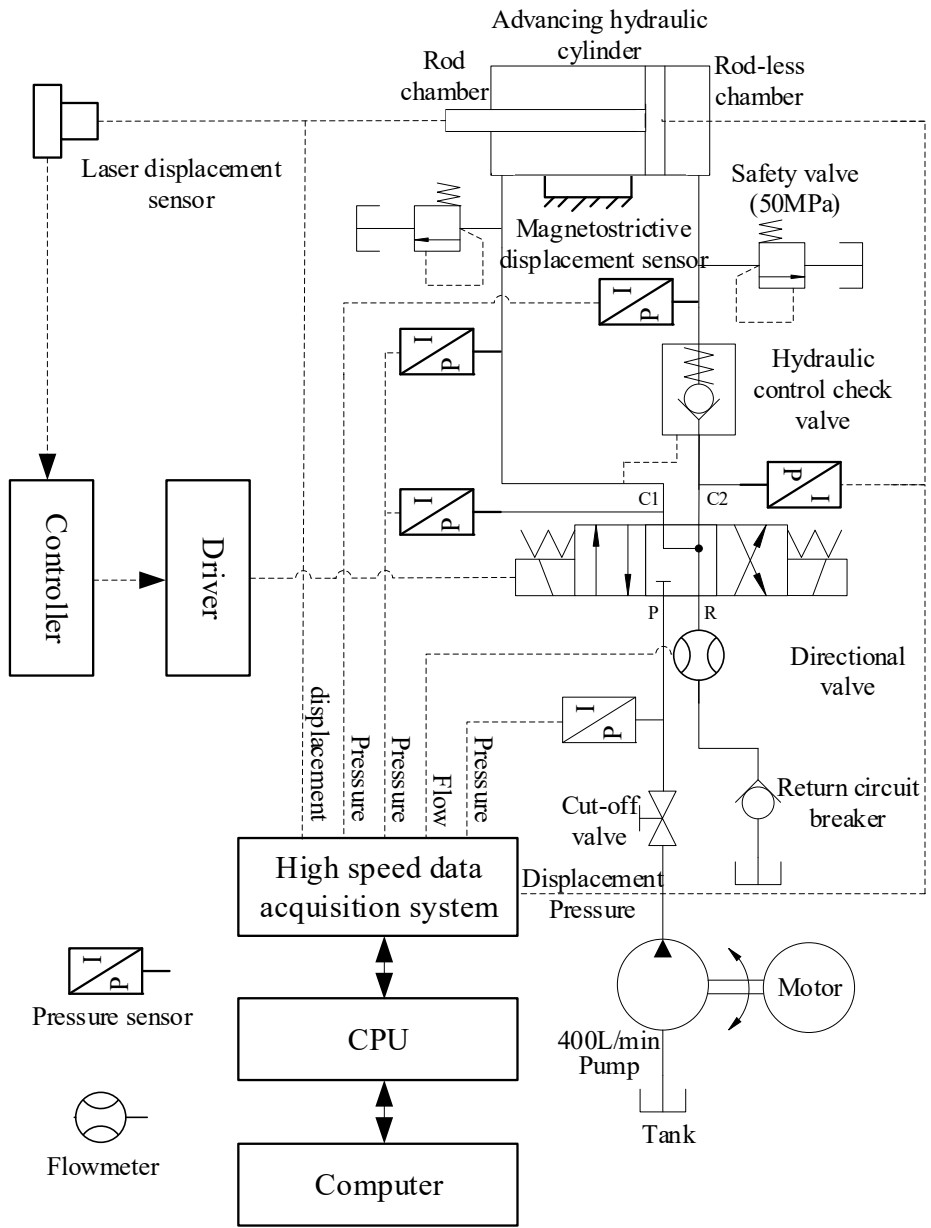

**Figure 3.** Hydraulic schematic diagram of the advancing test system.

Then, 40 MPa pressure sensors are connected at the inlet of the directional valve, the outlet of the directional valve port C1, the outlet of the directional valve port C2, the rod chamber of the cylinder, and the rod-less chamber of the cylinder to detect the pressure

changes when the advancing hydraulic cylinder reaches to different positions. A flowmeter is set at the return of the directional valve to measure the flow of the system in real time, a 1.5 m range laser displacement sensor is set in the front of the advancing hydraulic cylinder to measure the displacement of the rod and carry out closed loop control of the rod position, and a magnetostrictive displacement sensor is installed inside the advancing hydraulic cylinder to compare the collected data with the laser displacement sensor. The data acquisition frequency of the test system is 2 kHz, i.e., a group of pressure, flow, and displacement data are collected every 0.5 ms. The parameters of test equipment are shown in Table 2, and the connection diagram of experimental equipment is shown in Figure 4.

**Table 2.** Parameters of experimental equipment.

| Equipment | Parameter | Model |
|---|---|---|
| Emulsion pump | Nominal flow 400 L/min, nominal pressure 31.5 MPa | BRW400/37.5 |
| Pressure sensor | Measurement range 0–40 MPa, measurement accuracy 0.5% | KYB18G10M1PXCX-II |
| Return circuit breaker | Nominal flow 400 L/min, nominal pressure 16 MPa | FD400/16 |
| Directional valve | Nominal flow 400 L/min, nominal pressure 31.5 MPa | FHS400/31.5 |
| Hydraulic control check valve | Nominal flow 400 L/min, nominal pressure 50 MP a | FDYA400/31.5 |
| Magnetostrictive displacement sensor | Measurement range 900 mm, precision 0.1% | TMDXC(GUC1200/900/L).00 |
| Laser displacement sensor | Measurement 1600 mm, precision $\pm 1$ mm | DAN-10-150 |
| Advancing hydraulic cylinder | Cylinder/rod diameter 180/120 mm, stroke 900 mm | TMQTC(100/70*440)SL |
| Flowmeter | Nominal flow 400 L/min, precision 0.5% | FD-Q 50C |

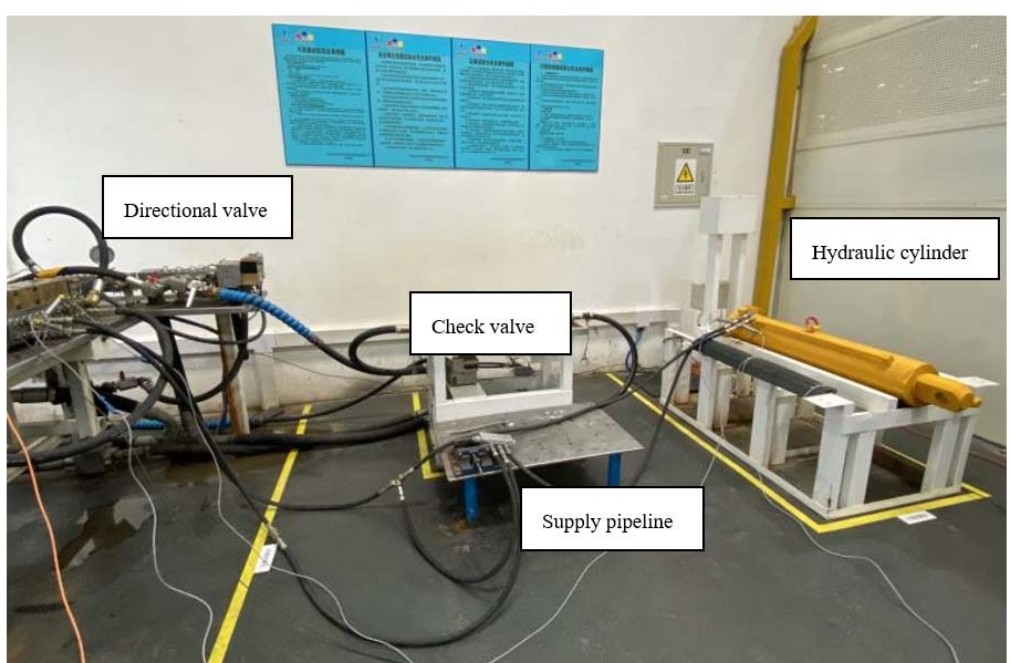

**Figure 4.** Connection diagram of the hydraulic system of advancing hydraulic cylinder.

### 3.3. Experimental Method

By adjusting the overflow valve of the emulsion pump to change the system pressure, the pressure and flow data at different positions are collected under the system pressure of 5, 10, 15, 20, 25, and 30 MPa, respectively, so as to find out the relationship between flow and pressure. In order to verify the data patterns of pressure, flow, and displacement under

different pipeline specifications, the data under DN10 and DN20 diameter are collected, respectively. Figure 5 shows the data under mean filtering method of the full stroke of the advancing cylinder when the system pressure is 5 MPa with DN20 pipeline size. The analysis shows that, when the hydraulic cylinder extends and retracts, the system flow fluctuates greatly due to the short time, so as to the correlation between flow and displacement of the cylinder is weak. This is mainly due to the delay of data acquisition by the flow sensor. The flow sensor cannot reflect the transient values, such as displacement and pressure sensors. It can only measure the average flow by adjusting the response time. If the response time is short, the measurement error will be large. In this experiment, the response time sets to 1 s, i.e., the sensor outputs the data by calculating the average flow value within 1 s. The small delay leads to a certain deviation of the flow data in the initial stage and then tends to be stable. In order to verify the accuracy of the flow data after stabilization, calculate the actual flow according to Formula (31). It can be seen from Figure 5 that, when the hydraulic cylinder is retracted under system pressure 5 MPa, the calculated flow is 72.32 L/min, while the flow curve in Figure 5 shows a value 75 L/min. The relative error between the actual flow and the sensor detecting is within 5%.

$$Q = A \cdot L \times 60{,}000 \ , \tag{31}$$

where: $L$—hydraulic cylinder displacement, m;

      $t$—hydraulic cylinder action time, s;

      $A$—hydraulic cylinder high-pressure side action area, m$^2$.

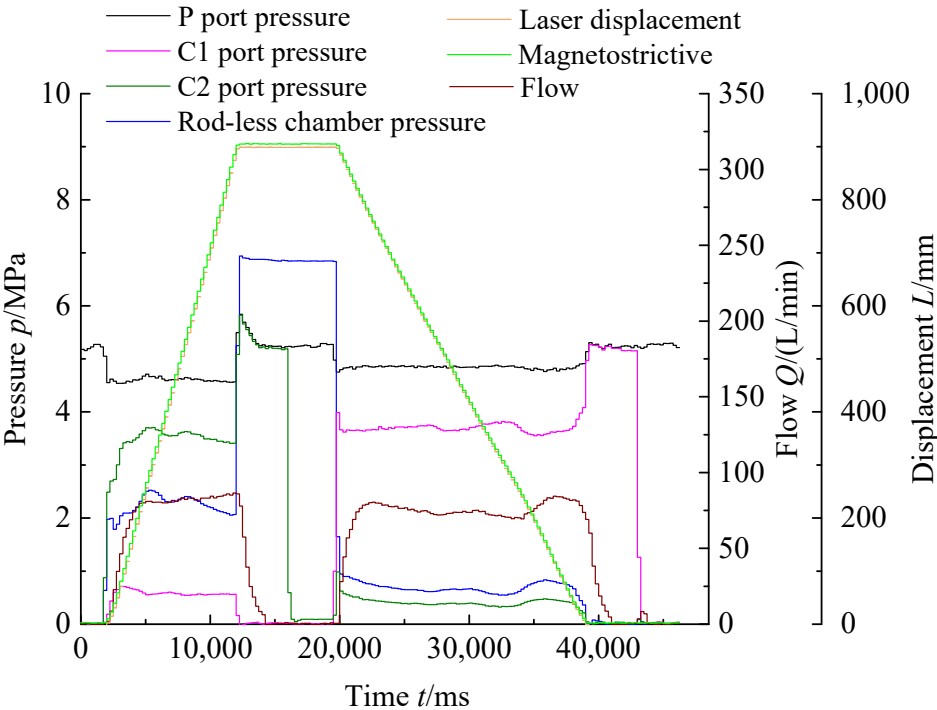

**Figure 5.** Full stroke mean filtering data of DN20 supply pipeline under 5 MPa system pressure.

## 4. Result and Discussion

### 4.1. Simulation Verification

In order to verify the effectiveness of the simulation model, a hydraulic system detection platform is set up. The comparison diagrams of displacement, pressure and flow between test and simulation of cyclic action of the hydraulic cylinder are shown in Figure 6.

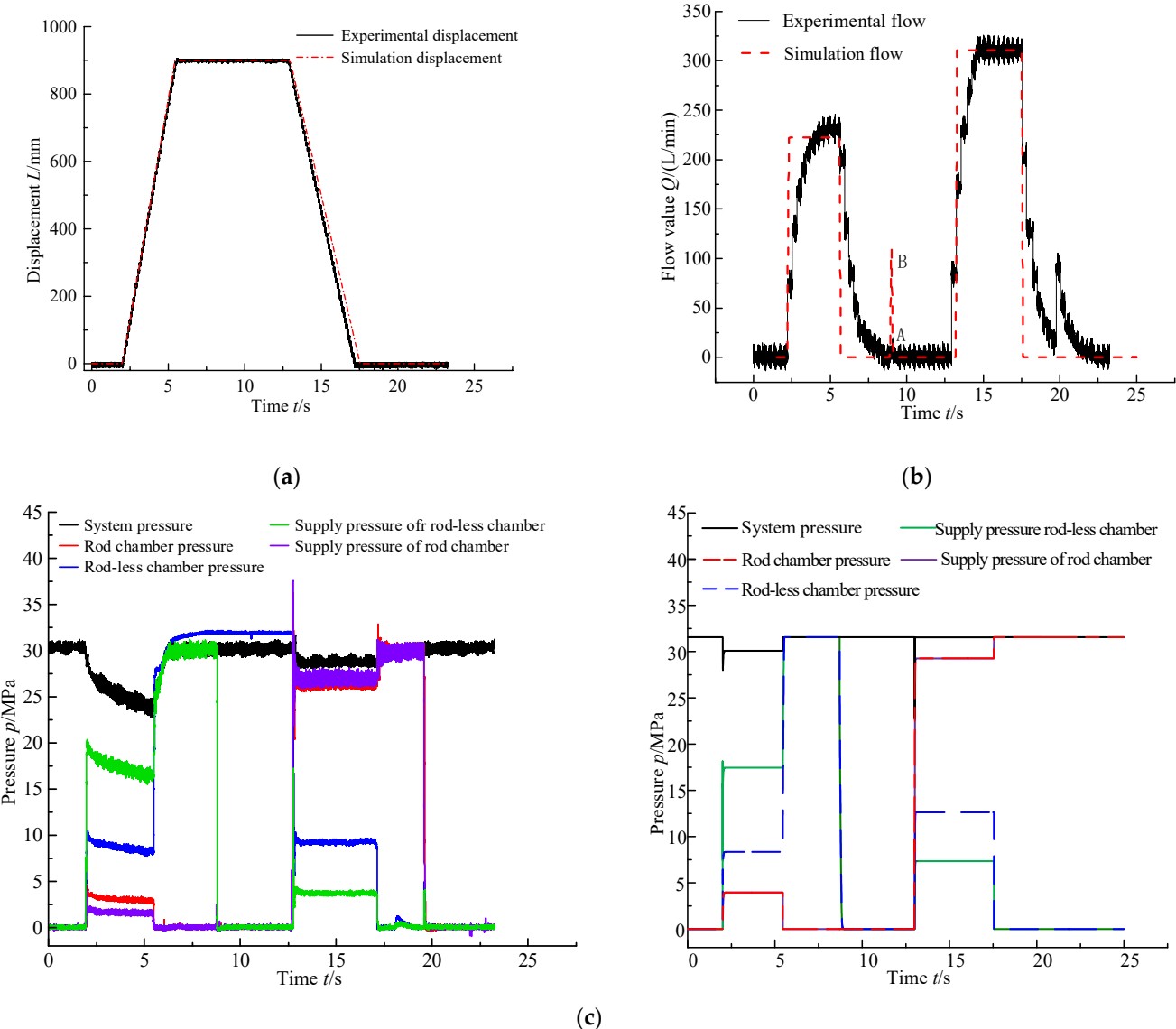

**Figure 6.** Comparison of the test and simulation data under 31.5 MPa system pressure: (**a**) Comparison diagram of experiment and simulation displacement; (**b**) comparison diagram of experiment and simulation outlet flow of electro-hydraulic control directional valve; (**c**) comparison diagram of experiment and simulation pressure.

The pressure of the hydraulic system is 31.5 MPa, and the liquid supply flow of the system is 400 L/min. The hydraulic cylinder extends and retracts without load, as shown in Figure 6a.

The test displacement and simulation curve of the hydraulic cylinder have good consistency. As is shown in Figure 6b, the pressure data of nodes of different components are consistent with the dynamic characteristics of simulation data. The main difference is caused by nonlinear factors (friction and elastic modulus of the system). When time *t* = 9 s, the cylinder extends to the maximum position. The directional valve is located in the middle position, and connects with the system return port. The hydraulic check valve locks cylinder rod-less chamber. At this time, the high-pressure liquid in the pipeline between the directional valve and the check valve flows to the system return port through the directional valve, causing the instantaneous change of flow, corresponding to point A of the test curve and point B of the simulation curve.

Figure 6c shows the pressure test curve and simulation curve of the extension and retraction of the cylinder. It can be seen from the experimental curve that the cylinder starts to extend at time equal to 2 s. Due to the static friction is much larger than the dynamic friction at the initial stage of start-up, the high-pressure fluid enters the rod-less chamber instantaneously, and other factors, the pressure in the rod-less chamber has a slow and slight drop at the initial stage. However, the simulation curve has the same static friction value as the dynamic friction value, resulting in stable pressure at the moment of start-up.

In conclusion, the AMESim simulation model of the advancing circuit hydraulic system is consistent with the actual system.

### 4.2. Simulation Result Analysis

#### 4.2.1. Relationship between Pressure Drop of Different Components and System Flow

When the hydraulic connecting pipes of the system are DN10 and DN 20, respectively, and the flow at the liquid supply port of the directional valve increase from 0 to 500 L/min linearly within 25 s, we obtain the relationship of the square root of the pressure drop and the flow rate between different components when the hydraulic cylinder extends, as shown in Figure 7a,b.

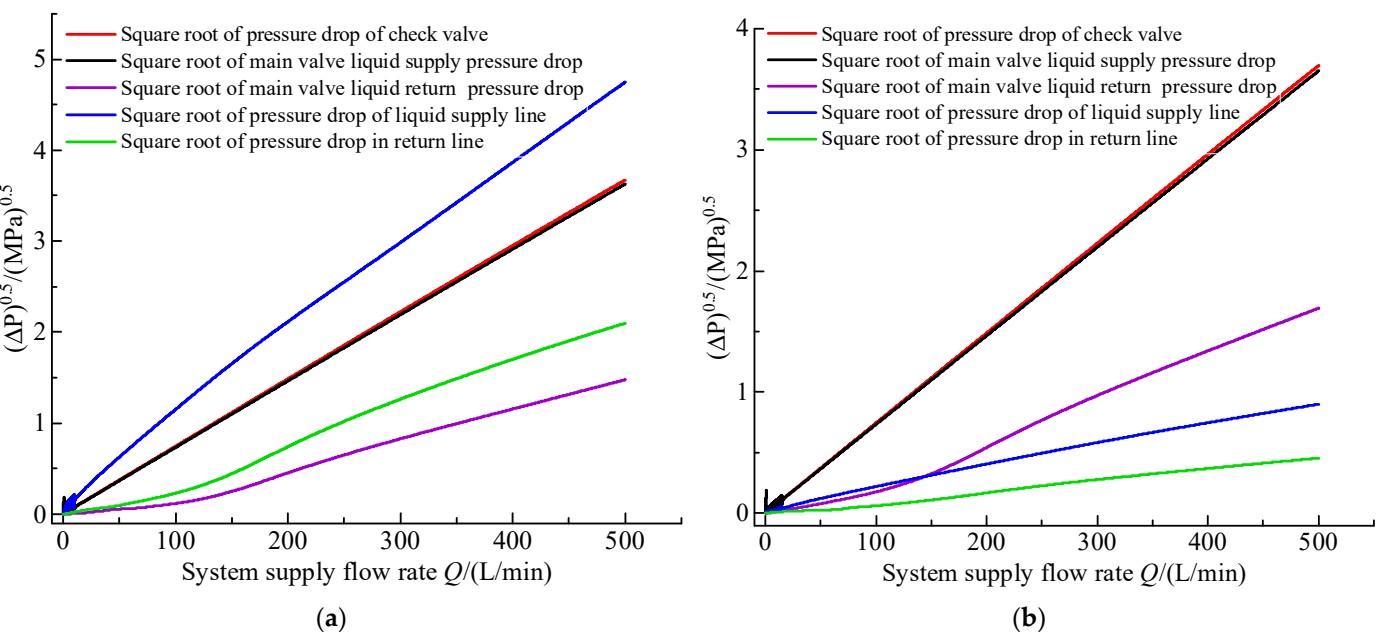

**Figure 7.** Relationship between liquid supply flow and pressure drop. (**a**) Relationship between liquid supply flow of DN10 pipeline and square root of pressure drop; (**b**) relationship between liquid supply flow of DN20 pipeline and square root of pressure drop.

The following conclusions are drawn by comparing Figure 7a,b: (1) with the increase of system liquid supply flow, the system flow has a linear relationship with the square root of pressure drop of directional valve and check valve, and a nearly linear relationship with pipeline; (2) the small diameter of DN10 pipeline leads to large pressure drop of inlet and outlet pipelines; and (3) different pipe diameters will also affect the pressure drop of other components of the system.

#### 4.2.2. Analysis of Pressure Drop Caused by Different Diameter of Liquid Supply Pipeline

In order to study the relationship between the square root of system pressure and flow under different supply pipeline diameters, the supply pipeline is changed from DN10 to DN40 for batch operations to analyze the influence of the diameter of the supply pipeline on the system parameters. The simulation results are shown in Figure 8.

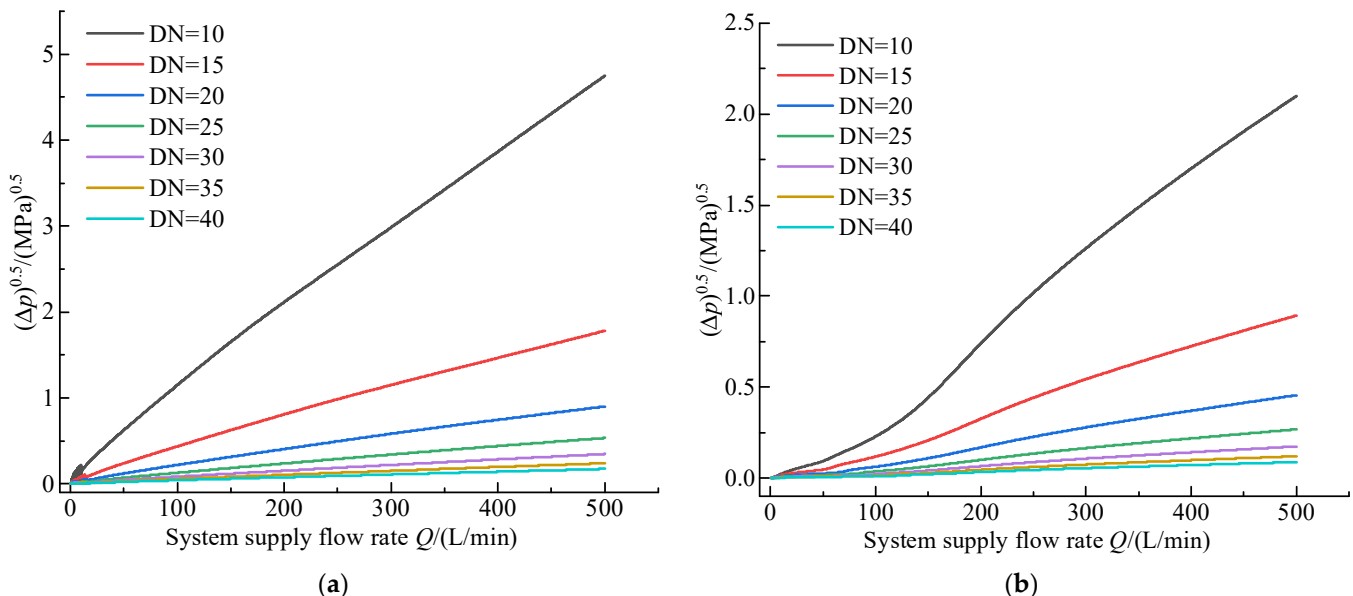

**Figure 8.** Relationship between supply flow and pressure drop of pipelines with different diameters: (**a**) Relationship between square root of pressure drop in inlet pipeline and supply flow; (**b**) relationship between square root of pressure drop in outlet pipeline and supply flow.

As can be seen from Figure 8a, when the system flow is less than 500 L/min, the pressure drop of the inlet pipeline decreases continuously with the increase of the diameter of the supply pipeline, and the square root of the pressure drop of the inlet pipeline is nearly linear with the supply flow. When the diameter of pipeline is above DN25, the pressure drop of the pipeline is low and has little impact in the system.

Since the outlet pipeline is directly connected with the main return, the pressure is low, so that the elastic modulus E changes greatly. It can be seen from Figure 8b that the system flow shows the approximately linear relationship with the square root of the pressure drop of the outlet pipeline.

### 4.2.3. Simulation Summary

The simulation shows that the flow of the hydraulic system has a linear relationship with the square root of the pressure drop of hydraulic components, pipelines, and joints, and its characteristics meet with the throttle flow equation of the valve port, so it can be equivalent to a fixed damping hole, which is also consistent with the formula results derived in the paper. The experimental platform and test data will be described in detail below.

### 4.3. Experimental Result Analysis

### 4.3.1. Relationship between Flow and Pressure Drop of Different Component

Analyzing the relationship between flow and pressure drop when the hydraulic cylinder extends and retracts, and we obtain the data between flow and pressure drop under different pump pressures, as shown in Tables 3 to 6.

**Table 3.** Hydraulic cylinder retraction data under DN10 pipeline diameter.

| System Pressure $p$ (MPa) | System Flow $Q$ (L/min) | Pressure Drop $\Delta p$ (MPa) | | |
|---|---|---|---|---|
| | | $\Delta p$ (Directional Valve) | $\Delta p$ (Pipeline) | $\Delta p$ (Check Valve) |
| 6 | 21.57 | 1.71 | 0.14 | 0.61 |
| 10.5 | 41.50 | 1.87 | 0.50 | 2.25 |
| 17 | 64.20 | 0.78 | 1.18 | 5.31 |
| 21.4 | 72.99 | 1.23 | 1.52 | 6.84 |
| 26.8 | 83.81 | 1.35 | 2.01 | 9.00 |
| 31 | 92.48 | 0.63 | 2.50 | 10.93 |

**Table 4.** Hydraulic cylinder extension data under DN10 pipeline diameter.

| System Pressure $p$ (MPa) | System Flow $Q$ (L/min) | Pressure Drop $\Delta p$ (MPa) | | |
|---|---|---|---|---|
| | | $\Delta p$ (Directional Valve) | $\Delta p$ (Pipeline) | $\Delta p$ (Check Valve) |
| 6 | 20.62 | 1.72 | 0.41 | 1.91 |
| 10.5 | 35.08 | 1.75 | 1.15 | 5.01 |
| 17 | 49.94 | 1.52 | 2.29 | 9.75 |
| 21.4 | 56.65 | 1.97 | 2.94 | 12.40 |
| 26.8 | 64.84 | 2.60 | 3.83 | 16.07 |
| 31 | 70.23 | 3.08 | 4.48 | 18.72 |

**Table 5.** Hydraulic cylinder retraction data under DN20 pipeline diameter.

| System Pressure $p$ (MPa) | System Flow $Q$ (L/min) | Pressure Drop $\Delta p$ (MPa) | | |
|---|---|---|---|---|
| | | $\Delta p$ (Directional Valve) | $\Delta p$ (Pipeline) | $\Delta p$ (Check Valve) |
| 5.2 | 71.22 | 1.15 | 0.15 | 0.28 |
| 11.5 | 179.74 | 0.91 | 0.97 | 1.79 |
| 16 | 222.15 | 0.90 | 1.48 | 2.73 |
| 22 | 266.95 | 1.30 | 2.14 | 3.95 |
| 25.8 | 291.09 | 1.51 | 2.54 | 4.67 |
| 30.6 | 316.37 | 1.77 | 3.00 | 5.48 |

**Table 6.** Hydraulic cylinder extension data under DN20 pipeline diameter.

| System Pressure $p$ (MPa) | System Flow $Q$ (L/min) | Pressure Drop $\Delta p$ (MPa) | | |
|---|---|---|---|---|
| | | $\Delta p$ (Directional Valve) | $\Delta p$ (Pipeline) | $\Delta p$ (Check Valve) |
| 5.2 | 77.61 | 1.08 | 0.18 | 1.26 |
| 11.5 | 134.60 | 3.09 | 0.54 | 3.36 |
| 16 | 159.10 | 4.28 | 0.76 | 4.67 |
| 22 | 187.64 | 5.95 | 1.05 | 6.44 |
| 25.8 | 202.95 | 6.91 | 1.23 | 7.58 |
| 30.6 | 216.28 | 7.83 | 1.40 | 8.60 |

According to the data in the table, we draw the relationship curve between flow and differential pressure. Figure 9 shows the statistical diagram of the extended and retracted flow and the square root of the differential pressure of the hydraulic cylinder under DN10 pipeline. Figure 10 shows the statistical diagram of the extended and retracted flow and the square root of the differential pressure of the hydraulic cylinder under DN20 pipeline. The analysis shows that, under different pipeline diameters, the system flow is linear with the square root of pipeline, check valve, and directional valve flow resistance. When the pipeline diameter is DN10, the system flow is low due to excessive flow resistance under the

same system pressure, and the maximum flow is less than 100 L/min. When the pipeline diameter is DN20, the system flow is large due to small flow resistance under the same system pressure, and the maximum flow is greater than 300 L/min.

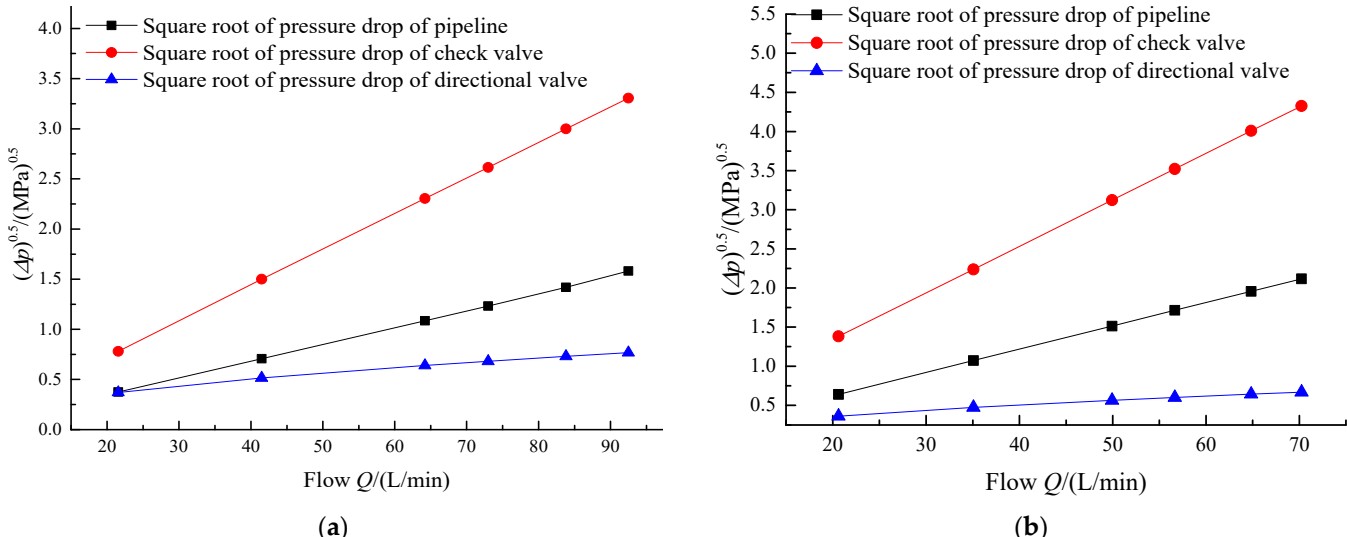

(**a**)                                          (**b**)

**Figure 9.** The relationship between flow and square root of differential pressure at all levels when the hydraulic cylinder is operated under DN10 pipeline diameter: (**a**) retracted; (**b**) extended.

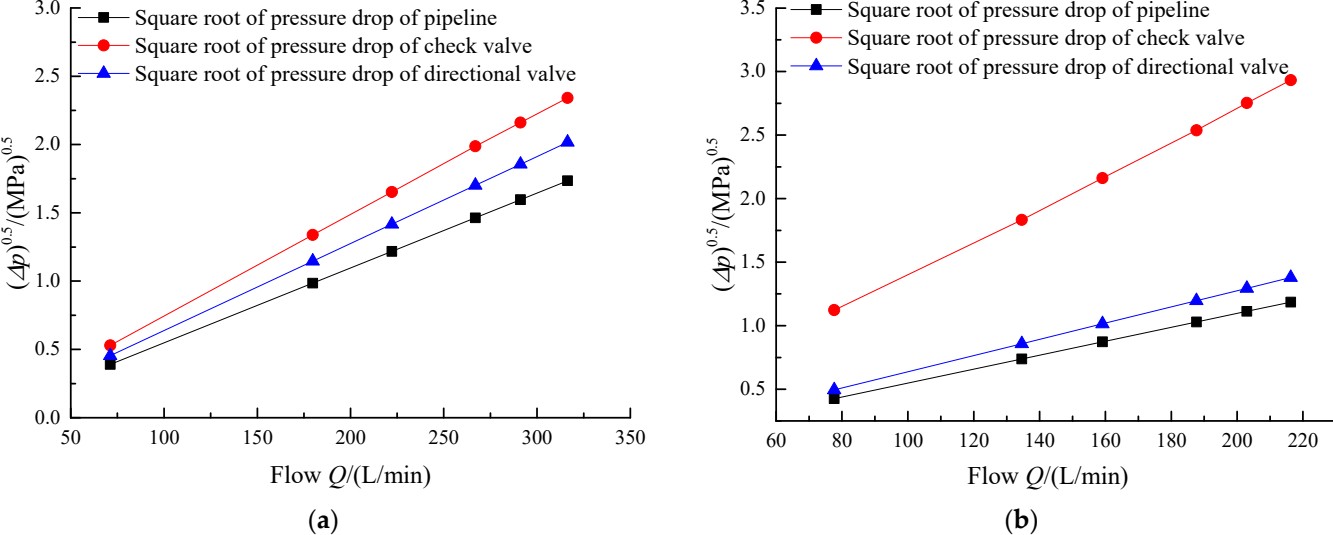

(**a**)                                          (**b**)

**Figure 10.** The relationship between flow and square root of differential pressure at all levels when hydraulic cylinder is operated under DN20 pipeline diameter: (**a**) retracted; (**b**) extended.

### 4.3.2. Relationship between Flow and Inlet and Outlet Pressure

In order to verify the relationship between system flow and inlet and outlet pressure under different pipeline conditions, the working state parameter tables under different working conditions are summarized, and the system parameters are shown in Table 7. According to formulas (16) and (28), the system flow is taken as the horizontal ordinate, and the output function $f(p_{in}, p_{out})$ (includes $f_s(p_{in}, p_{out})$ and $f_h(p_{in}, p_{out})$) is taken as the vertical ordinate based on the input under different working states. The experimental results are shown in Figures 11 and 12.

**Table 7.** System parameters table under different liquid supply conditions.

| | Hydraulic Cylinder Extends | | | | Hydraulic Cylinder Retracts | | | |
|---|---|---|---|---|---|---|---|---|
| | Flow $Q$ (L/min) | Inlet Pressure $p_{in}$ (MPa) | Return Pressure $p_{out}$ (MPa) | $f_s$ $(p_{in}, p_{out})$ (N)$^{0.5}$ | Flow $Q$ (L/min) | Inlet Pressure $p_{in}$ (MPa) | Return Pressure $p_{out}$ (MPa) | $f_h$ $(p_{in}, p_{out})$ (N)$^{0.5}$ |
| DN10 pipeline | 20.62 | 6 | 0.62 | 0.121 | 21.57 | 6 | 0.62 | 0.073 |
| | 35.08 | 10.5 | 0.74 | 0.162 | 41.5 | 10.5 | 0.53 | 0.103 |
| | 49.94 | 17 | 0.82 | 0.207 | 64.2 | 17 | 0.53 | 0.135 |
| | 56.65 | 21.4 | 0.89 | 0.233 | 72.99 | 21.4 | 0.54 | 0.152 |
| | 64.84 | 26.8 | 0.83 | 0.261 | 83.81 | 26.8 | 0.57 | 0.171 |
| | 70.23 | 31 | 0.91 | 0.281 | 92.48 | 31 | 0.65 | 0.184 |
| DN20 pipeline | 77.61 | 5.2 | 0.75 | 0.112 | 71.22 | 5.2 | 0.37 | 0.071 |
| | 134.60 | 11.5 | 0.65 | 0.170 | 179.74 | 11.5 | 1.18 | 0.101 |
| | 159.10 | 16 | 0.91 | 0.201 | 222.15 | 16 | 1.84 | 0.117 |
| | 187.64 | 22 | 1.22 | 0.235 | 266.95 | 22 | 2.63 | 0.136 |
| | 202.95 | 25.8 | 1.46 | 0.255 | 291.09 | 25.8 | 3.19 | 0.146 |
| | 216.28 | 30.6 | 1.67 | 0.277 | 316.37 | 30.6 | 3.67 | 0.160 |

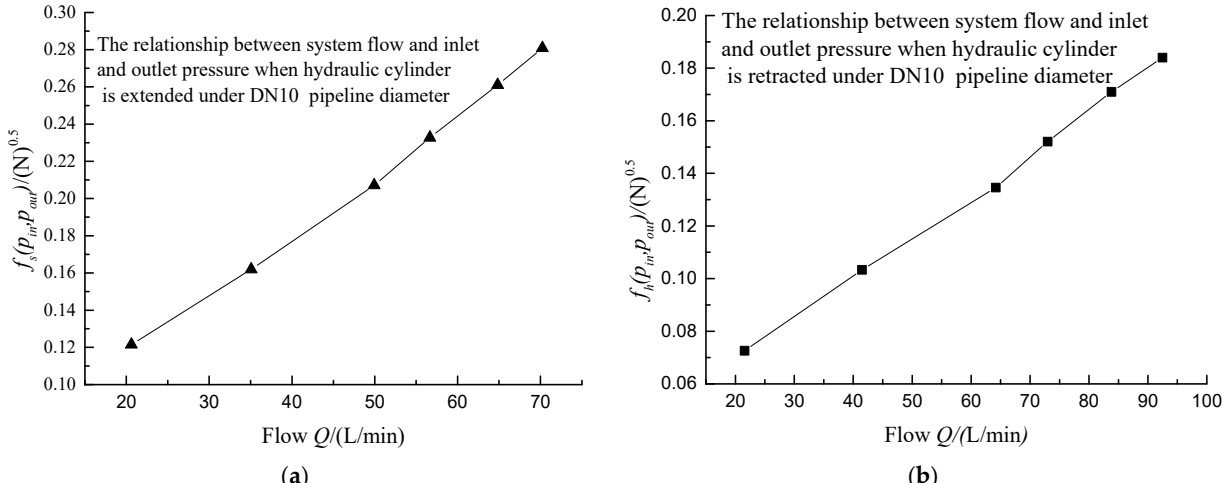

**Figure 11.** Relationship between flow and inlet and outlet pressure of hydraulic cylinder under DN10 diameter: (**a**) extended; (**b**) retracted.

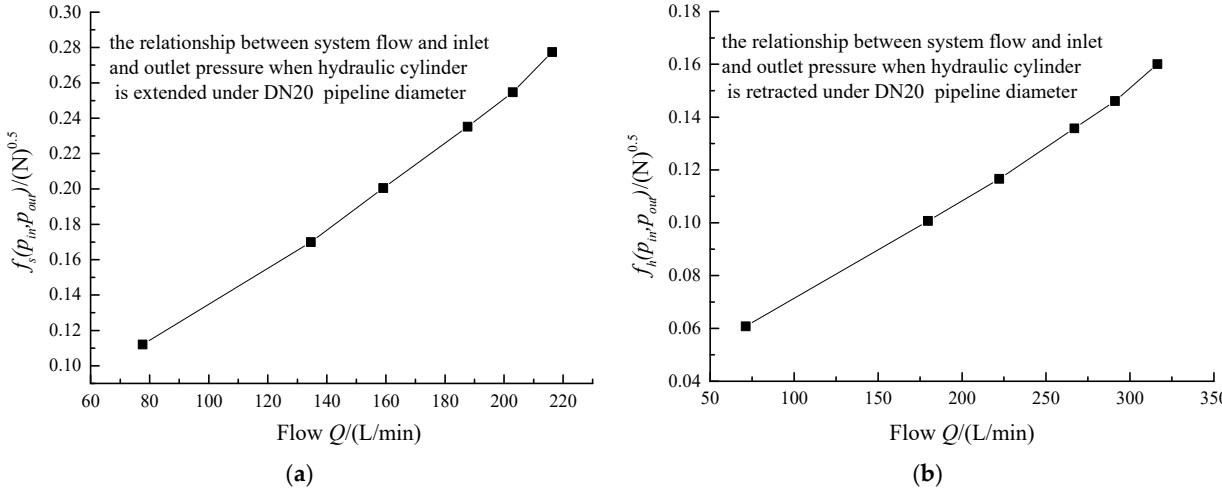

**Figure 12.** Relationship between flow and inlet and outlet pressure of hydraulic cylinder under DN20 diameter: (**a**) extended; (**b**) retracted.

By analyzing Figures 11 and 12, the system flow under different pipeline conditions is basically linear, with the function $f(p_{in}, p_{out})$ based on the inlet and outlet pressure as the variable. The linearity error of some low pressure areas and high pressure areas is mainly due to the change of elastic modulus in high and low pressure areas, friction, and other nonlinear loads, but the load has a small proportion in the system calibration or is equivalent to a linear variable, so the influence on the final calculation result can be ignored.

### 4.4. Fitting Verification

Next, assign values to Equations (15) and (27) according to the system structural parameters to calculate the fitting flow of different pipeline layouts and working states, statistically analyze the fitted flow and the actual flow, and calculate the relative error according to Equation (32). The statistical results are shown in Table 8, and the fitting relationship diagram is shown in Figure 13.

$$\delta = \frac{|Q_n - Q_s|}{Q_s} \times 100\%, \tag{32}$$

where $Q_n$—fitting flow; $Q_s$—measured flow.

**Table 8.** Comparison table of fitting flow and measured flow under different pipeline diameters.

| | System Pressure (MPa) | 5 | 10 | 15 | 20 | 25 | 30 |
|---|---|---|---|---|---|---|---|
| Hydraulic cylinder extends under DN10 pipeline | Actual flow (L/min) | 20.62 | 35.08 | 49.94 | 56.65 | 64.84 | 70.23 |
| | Fitting flow (L/min) | 23.16 | 39.04 | 49.89 | 56.15 | 62.90 | 67.72 |
| | Relative error (%) | 12.32 | 11.29 | 0.11 | 0.88 | 2.99 | 3.57 |
| Hydraulic cylinder retracts under DN10 pipeline | Actual flow (L/min) | 21.57 | 41.5 | 64.2 | 72.99 | 83.81 | 92.48 |
| | Fitting flow (L/min) | 25.894 | 47.234 | 64.53 | 72.656 | 81.738 | 87.952 |
| | Relative error (%) | 20.05 | 13.82 | 0.51 | 0.46 | 2.47 | 4.90 |
| Hydraulic cylinder extends under DN20 pipeline | Actual flow (L/min) | 77.61 | 134.60 | 159.10 | 187.64 | 202.95 | 216.28 |
| | Fitting flow (L/min) | 88.70 | 134.64 | 159.19 | 186.12 | 201.96 | 219.38 |
| | Relative error (%) | 14.29 | 0.03 | 0.06 | 0.81 | 0.49 | 1.44 |
| Hydraulic cylinder retracts under DN20 pipeline | Actual flow (L/min) | 71.22 | 179.74 | 222.15 | 266.95 | 291.09 | 316.37 |
| | Fitting flow (L/min) | 79.83 | 191.80 | 222.18 | 258.26 | 277.25 | 303.84 |
| | Relative error (%) | 12.09 | 6.71 | 0.01 | 3.25 | 4.75 | 3.96 |

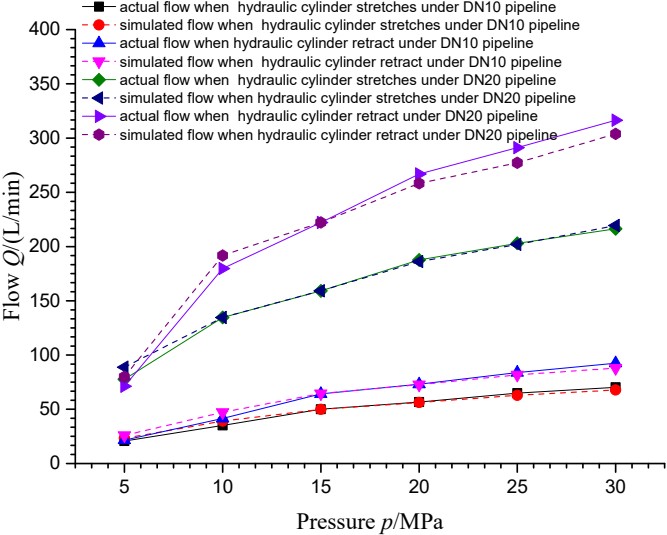

**Figure 13.** Comparison diagram of actual flow and fitting flow under different pipeline parameters and working conditions.

As can be seen in Figure 13 and Table 8, the coincidence degree of actual flow and fitting flow under different pipeline parameters and working conditions is good. Under the conditions of DN10 and DN20 pipelines, when the pressure is above 10 MPa, the system flow coincidence error is within 5%, and, under the conditions of 5–10 MPa, the system flow coincidence error is within 13%, which is mainly caused by the nonlinearity and large variation of the elastic model in the low pressure region. The elastic modulus is equivalent to a constant when system pressure is above 10 MPa.

## 5. Conclusions

The pressure and flow coupling-based precise position control was proposed for the hydraulic cylinder in the hydraulic support. The flow and stroke control model was established based on the flow continuity equation and Newton Euler dynamic equation. The hydraulic system simulation software AMESim proved the effectiveness and correctness of the control model. In addition, our proposed test system demonstrated that, when the system pressure was larger than 10 MPa, the error between the data determined by the fitting algorithm and the actual detection data was within 5%. In the future, in order to improve the accuracy and promote the application of position control based on pressure detection, we will verify the transient effects of other factors, such as load, volume, and delay on cylinder position control, as well as propose a transient control model based on volume compression formula, so as to finally realize the unification of transient model and steady-state model.

**Author Contributions:** Conceptualization, R.Z.; methodology, R.Z.; software, Z.Q.; validation, L.M.; formal analysis, X.Y.; investigation, L.M. and Z.Q.; resources, R.Z.; data curation, R.Z. and L.M.; writing—original draft preparation, R.Z.; writing—review and editing, L.M. and Z.Q.; supervision, R.Z.; project administration, X.Y.; funding acquisition, R.Z. All authors have read and agreed to the published version of the manuscript.

**Funding:** This research was funded by China Coal Technology & Engineering Group, grant number 2020-TD-MS009.

**Institutional Review Board Statement:** Not applicable.

**Informed Consent Statement:** Not applicable.

**Data Availability Statement:** Not applicable.

**Acknowledgments:** We acknowledge the funding support from CCTEG (project account code: 2020-TD-MS009) for this research and support of the Laboratory of Beijing Tianma Intelligent Control Technology Co., Ltd.

**Conflicts of Interest:** The authors declare no conflict of interest.

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
