# Peer review of "Research and Experimental Analysis of Hydraulic Cylinder Position Control Mechanism Based on Pressure Detection"

_machines, doi:10.3390/machines10010001_

Round 1

Reviewer 1 Report

The numerical and experimental simulation methodologies are well described. The simulation and experimental results are correctly presented and analyzed. However, the application to the cylinder position control system can clearly be improved.

1. Specify how the results obtained allow the control model they propose to be effective

2. it is necessary to demonstrate more clearly the effectiveness of the proposed model

Author Response

Point 1: Specify how the results obtained allow the control model they propose to be effective

Response1: Thank you for your comments. Based on the composition of the hydraulic system, the input flow control model with input and output pressure as variables under steady-state conditions is established. Finally, by substituting the parameters of system parts into the control model and comparing the actual data, the error between system flow and fitting flow is 5% when the system pressure is larger than 10MPa, which verifies the feasibility of replacing actual flow by fitting flow. Since the working pressure of the hydraulic system of the fully mechanized mining face is above 10MPa in most of scenarios, the calculation model has good adaptability. Finally, through the real-time integration of the fitting flow, the estimated position of the hydraulic cylinder can be obtained, so as to finally determine the displacement of the hydraulic support equipment.

This paper mainly analyzes the position control principle based on input and output pressure under steady-state conditions. When the system pressure is below 10MPa, the model and actual errors are large because of the elastic modulus and the proportion of system friction cannot be ignored, which is also the next step we will continue to study. Through the experimental study of piece-wise linearization of nonlinear factors under 10MPa, it is ensured that the flow fitting is matched with the actual data under low pressure, and the error is within 5%.

In the next step, we will also carry out flow calculation and analysis during transient conditions. The influence of system input and output pressure on position control under unbalanced condition is analyzed in order to realize the integration and unification of relevant theories, so as to achieve industrial test and applications.

Point 2: It is necessary to demonstrate more clearly the effectiveness of the proposed model

Response2: Thank you for your comments. This paper verifies the application scenario of the hydraulic system of fully mechanized mining face. Through simulation, numerical analysis and experimental research, it is concluded that when the system pressure is more than 10MPa, the error between the theoretical model and the experiment is less than 5%, which meet the application requirements of the working face. Since large error and unclear mechanism between the control model and the actual results exist under low pressure, we will continue to conduct in-depth research in the future.

Reviewer 2 Report

This paper presents an experimental analysis of hydraulic cylinder position control mechanism based on pressure detection. This is an interesting topic.  The current manuscript should be discussed further.

  1. It is difficult to realize the contribution of this paper based on the abstract,  the introductions, and the conclusion. These sections should be presented rigorously.
  2. The introduction should be rewritten to show the motivation of this study. It is hard to realize the necessity of this study.
  3. The working principle of the hydraulic system in figure 1 should be described.
  4. Please explain how to obtain the equation (15).
  5. Please explain the differences between experimental and simulation results in figure 6, 7 and 8. What is the reason?

Author Response

Point 1: It is difficult to realize the contribution of this paper based on the abstract, the introductions, and the conclusion. These sections should be presented rigorously.

Response1: Thank you for your comments. Necessary modifications have been made in the abstract, the introductions and the conclusions. We add the urgency for realizing automatic following of hydraulic support in fully mechanized mining face. Through simulation, numerical analysis and experimental research, the estimated position of the hydraulic cylinder can be obtained from system pressure, so as to finally determine the displacement of the hydraulic support equipment.

Point 2: The introduction should be rewritten to show the motivation of this study. It is hard to realize the necessity of this study.

Response2: Thank you for your comments. Please see the content added in the introduction part marked red. The introduction has been revised. The composition and working characteristics of the current hydraulic system of fully mechanized mining face are added. the difficulties and requirements of cluster hydraulic cylinder control at present are illustrated. Based on these challenges, the significance and purpose of studying position control based on pressure detection are described.

Point 3: The working principle of the hydraulic system in figure 1 should be described.

Response3: Thank you for your comments. Relevant description has been added for figure 1.

Point 4: Please explain how to obtain the equation (15).

Response4: Thank you for your comments. Equation 2 is corrected, please see the attached file for detail information.

For equation 2, when hydraulic cylinder extends, F take minus sign; when hydraulic cylinder retracts, F take plus sign.

From all equations above, we can derive equation (15)

Point 5: Please explain the differences between experimental and simulation results in figure 6, 7 and 8. What is the reason?

Response5: Thank you for your comments. Since the structure of the article was modified last time, Figure 7 (a) was incorrectly replaced. Now Figure 7 (a) is corrected. Please see the red fonts in the according places.

Figure 6 shows the simulation and experiment results comparison of extending and retracting under DN10 pipeline. The data matches well between the simulation model and the experiment system.

Figure 7 shows the analysis of the relationship between the system flow and the pressure difference of hydraulic units under DN10 and DN20 pipe diameters on the basis of the feasibility of the simulation model in Figure 6.

Figure 8 analyzes the relationship between the system flow and the pressure difference of DN10-DN40 pipe diameters, verifies that the flow basically has a linear relationship with the pipeline flow resistance, and verifies the consistency of the control model and deduced formula. It can be noticed that in the end of each curve, the y-axis flattens out. This sis because we add a relieve valve in the simulation mode, which will open when system pressure is above 31.5MPa. As system flow increases, the pressure drop builds in the system. When the pressure builds up to 31.5MPa, the flow returns to the tank through bypass, which is the reason that as flow rate increases, the system pressure maintains.

Reviewer 3 Report

The presented article presents the mathematical model and the results of pressure loss tests in the hydraulic supply system of hydraulic cylinders used in mining supports. The work is presented as part of a larger scientific research venture aimed at developing an automatic system of mining supports, which is carried out independently in many countries. However, the article focuses only on determining the pressure drops in the presented system. In this regard, the following substantive comments to the article were formulated:
• The authors used a turbulent flow model for the description of the flow type, although it was not directly proved in the article that this model is adequate for the entire range of tested hydraulic line cross-sections (no Reynolds number was determined for individual cases).
• The following errors were noticed in Figure 3:
o the hydraulic pump is graphically showed as the engine,
o the shut-off valve downstream of the pump is marked as if it is adjustable,
o the non-return valve located on the drain line is set in a shut-off position, which prevents the system from operating properly,
o the three-position valve has no spring or lock markings,
o the maximum valve located on the supply line of the piston rod chamber of the actuator is controlled from the side of the tank.
• In Figure 5, the timeline is probably wrong describes (the current description shows that the time of the extension stroke is almost 3 hours).
• Figures 7 and 8 show the graphs of changes in the pressure difference, which show that from a certain flow rate its further increase does not change the value of the flow resistance, which is directly contrary to the formula No. 8 and the test results.
• In formula No. 28, there is the fh parameter, while in the description preceding the table No. 7, there is the f parameter, and in the table itself, the fs parameter, which seems incomprehensible.
• 8 graphs are described in figure 8, but only 6 are shown.
• Comments on the mathematical model may be of significant importance in the interpretation of the summary results collected in Table 8 and Figure 13.
The authors put excessive spaces in the name of (Longwall Automation Steering Committee) in line 75 of the text.

Author Response

Point 1: The authors used a turbulent flow model for the description of the flow type, although it was not directly proved in the article that this model is adequate for the entire range of tested hydraulic line cross-sections (no Reynolds number was determined for individual cases).

Response1: Thank you for your comments. Because most of applications of hydraulic systems in fully mechanized mining face under large flow scenario, based on journal Study on coupling mechanism of pressure and flow in following hydraulic system of mining face, generally, Reynolds number Re>>2300. Therefore, the flow state is turbulent. According to formula

Where, u-flow velocity, m/s;

D-hydraulic diameter, m;

v-kinematic viscosity, m2/s, emulsion is 8.9e-7.

The Reynolds number of pipe diameter DN10-dn50 under the condition of minimum flow of 10L/min shall be determined according to the service conditions. As can be seen in the table, the calculated Reynolds number under all working conditions is greater than 2300, which shows that the working condition is turbulent.

Q

DN10

DN20

DN30

DN40

DN50

10L/min

23843.44

11921.72

7947.812

5960.859

4768.687

The calculated Reynolds number under different pipe diameters in the range of 10-100L/min is plotted, as shown in the following figure, please see the figure in the attachment.

Point 2:The following errors were noticed in Figure 3:

  • the hydraulic pump is graphically showed as the engine,
  • the shut-off valve downstream of the pump is marked as if it is adjustable,
  • the non-return valve located on the drain line is set in a shut-off position, which prevents the system from operating properly,
  • the three-position valve has no spring or lock markings,
  • the maximum valve located on the supply line of the piston rod chamber of the actuator is controlled from the side of the tank.

Response2: Thank you for your comments. â‘ -â‘£ errors have been corrected.

For ⑤ statement, the value we use in the experiment is Y Solenoid valve, the hydraulic diagram is showing below. The working ports C1 and C2 connect with the tank when the solenoid is in the middle. Please see the figure in the attachment.

Point 3: In Figure 5, the timeline is probably wrong describes (the current description shows that the time of the extension stroke is almost 3 hours).

Response3: Thank you for your comments. Time unit is corrected to ms.

Point 4: Figures 7 and 8 show the graphs of changes in the pressure difference, which show that from a certain flow rate its further increase does not change the value of the flow resistance, which is directly contrary to the formula No. 8 and the test results.

Response4: Thank you for your comments. Since the structure of the article was modified last time, Figure 7 (a) was incorrectly replaced. Now Figure 7 (a) is corrected. Please see the red fonts in the according places.

As can be seen in figure 2, the simulation model has a relieve valve, which will open when system pressure is above 31.5MPa. As system flow increases, the pressure drop builds in the system. When the pressure builds up to 31.5MPa, the flow returns to the tank through bypass, which is the reason that as flow rate increases, the system pressure maintains.

Point 5: In formula No. 28, there is the fh parameter, while in the description preceding the table No. 7, there is the f parameter, and in the table itself, the fs parameter, which seems incomprehensible.

Response5: Thank you for your comments. fs is in formula No.16, which is calculated when the hydraulic cylinder extends. fh is in formula No.28, which is calculated when the hydraulic cylinder retracts. f is the combination of fs and fh. All names of Y-axis have been changed in Figure 11 and 12 to according fs and fh.

Point 6: 8 graphs are described in figure 8, but only 6 are shown.

Response6: Thank you for your comments. I believe that you mean figure 13. One set of data was calculated incorrectly in table 8. The data and the figure are corrected now.

Point 7: Comments on the mathematical model may be of significant importance in the interpretation of the summary results collected in Table 8 and Figure 13.

Response7: Thank you for your comments. One set of data calculated in table 8 was wrong for Hydraulic cylinder retracts under DN10 pipeline, and now is corrected.

Point 8: The authors put excessive spaces in the name of (Longwall Automation Steering Committee) in line 75 of the text.

Response8: Thank you for your comments. Additional space in the name of (Longwall Automation Steering Committee) is deleted.

Round 2

Reviewer 2 Report

This revised manuscript has been improved significantly. However, some issues should be discussed further.

1. Equation 2 should be represented as the force direction in Figure 1.

2. The author should explain again the difference between the simulation and experimental results in Figure 6b and 6c.

3. There are too many paragraphs in the introduction. It should be improved more for readability. 

Author Response

Point 1: Equation 2 should be represented as the force direction in Figure 1.

Response1: Thank you for your comments. Equation 2 is changed as the force direction shown in the figure, and the direction of the joint force F is related to the direction of the friction and load. All other equations are changed as the load direction as in figure 1.

Point 2: The author should explain again the difference between the simulation and experimental results in Figure 6b and 6c.

Response2: Thank you for your comments. The difference between the simulation and experiment results is added above the figure in the article.

Point 3: There are too many paragraphs in the introduction. It should be improved more for readability.

Response3: Thank you for your comments. In order to ensure the readability and coherence, the structure and paragraphs of the introduction are adjusted. Paragraphs 1 and 2 are integrated into one paragraph, which mainly describes the main application scenarios of the current position control system.The integration of paragraphs 4 and 5 mainly illustrates the special needs and main difficulties of position control in the coal industry. Paragraph 6 is deleted, which is not well integrated with the system theme and does not affect the integrity of the overall structure.

Reviewer 3 Report

The following remarks were not clarified or corrected:
• The following errors were noticed in Figure 3:
o the hydraulic pump is graphically showed as the engine,
o the maximum valve located on the supply line of the piston rod chamber of the actuator is controlled from the side of the tank.
• Figures 7 and 8 show the graphs of changes in the pressure difference, which show that from a certain flow rate its further increase does not change the value of the flow resistance, which is directly contrary to the formula No. 8 and the test results.
• The authors put excessive spaces in the name of (Longwall Automation Steering Committee) in line 75 of the text.

Author Response

Point 1: The following errors were noticed in Figure 3:

o the hydraulic pump is graphically showed as the engine,

o the maximum valve located on the supply line of the piston rod chamber of the actuator is controlled from the side of the tank.

Response1: Thank you for your comments. All points are corrected in the figure.

Point 2:Figures 7 and 8 show the graphs of changes in the pressure difference, which show that from a certain flow rate its further increase does not change the value of the flow resistance, which is directly contrary to the formula No. 8 and the test results.

Response2: Thank you for your comments. Figure 7 and 8 are replaced by new simulation results, the flow rate and the root of the pressure are linear. More detailed responses are in the attachment.

Point 3: The authors put excessive spaces in the name of (Longwall Automation Steering Committee) in line 75 of the text.

Response3: Thank you for your comments. Additional space in the name of (Longwall Automation Steering Committee) is deleted.

Round 3

Reviewer 2 Report

No more comment.

Reviewer 3 Report

Thank you for considering and taking my comments into account.